# Cytotoxic and Pro-Apoptotic Effects of Leaves Extract of *Antiaris africana* Engler (*Moraceae*)

**DOI:** 10.3390/molecules27227723

**Published:** 2022-11-09

**Authors:** Khadidiatou Thiam, Minjie Zhao, Eric Marchioni, Christian D. Muller, Yerim M. Diop, Diane Julien-David, Fathi Emhemmed

**Affiliations:** 1Laboratoire de Chimie Analytique et Bromatologie, Département de Pharmacie, Université Cheikh Anta Diop de Dakar, Dakar 5005, Senegal; 2IPHC, UMR 7178 CNRS, Faculté de Pharmacie, Université de Strasbourg, 67401 Ilkirch, France

**Keywords:** *Antiaris africana* Engler, cytotoxicity, cardiac glycosides, human cancer cells line, apoptosis

## Abstract

*Antiaris africana* Engler leaves have been used in Senegalese folk medicine to treat breast cancer. The present study aimed to investigate the anticancer potential of *Antiaris africana* Engler leaves using several human cancer cell lines. The leaves of *Antiaris africana* Engler were extracted in parallel with water or 70% ethanol and each extract divided into three parts by successive liquid–liquid extraction with ethyl acetate and butanol. The phytochemical components of the active extract were investigated using ultra-performance liquid chromatography-diode array detector-quadrupole time-of-flight tandem mass spectrometry (UPLC-DAD-QTOF-MS/MS). The cytotoxic and cytostatic effects of each extract, as well as their fractions, were evaluated in vitro via flow and image cytometry on different human cancer phenotypes, such as breast (MCF-7), pancreas (AsPC-1), colon (SW-620) and acute monocytic leukemia (THP-1). Both hydro-alcoholic and aqueous extracts induced strong apoptosis in MCF-7 cells. The water fraction of the hydro-alcoholic extract was found to be the most active, suppressing the cell growth of MCF-7 in a dose-dependent manner. The half maximum effective concentration (EC_50_) of this fraction was 64.6 ± 13.7 μg/mL for MCF-7, with equivalent values for all tested phenotypes. In parallel, the apoptotic induction by this fraction resulted in a EC_50_ of 63.5 ± 1.8 μg/mL for MCF-7, with again equivalent values for all other cellular tested phenotypes. Analysis of this fraction by UPLC-DAD-QTOF-MS/MS led to the identification of hydroxycinnamates as major components, one rutin isomer, and three cardiac glycosides previously isolated from seeds and bark of *Antiaris africana* Engler and described as cytotoxic in human cancer models. These results provide supportive data for the use of *Antiaris africana* Engler leaves in Senegal.

## 1. Introduction

There were approximately 19.3 million new cancer cases and nearly 10 million cancer deaths worldwide in 2020 according to the American Cancer Society and the International Agency for Research on Cancer [1]. More than 600,000 patients die every year in Africa because of cancer [2]. Breast cancer is the most frequently diagnosed cancer and the leading cause of death [3]. This trend is expected to increase in the coming decades in African countries, due to the absence of a national cancer control program, the lack of specialized infrastructure and qualified human resources and the exorbitant cost of care.

Herbal medicines used as anti-cancer remedies have provided modern medicine with effective cytotoxic pharmaceuticals [4,5]. More than 60% of the currently available anti-cancer drugs are derived directly or indirectly from natural products, including vinblastine, vincristine, paclitaxel and irinotecan [3,6]. Traditional medicine mainly uses herbal therapies [7]. In Africa, this traditional medicine is sometimes the only source of affordable and accessible care, especially for the poorest patients. *Antiaris africana* Engler (Moraceae) is used in Cameroon folk medicine in the treatment of cancer [2]. Bark extracts of *Antiaris africana* Engler are used for the treatment of chest pain and to relieve rheumatic, respiratory and stomach pain in Nigeria [8]. Leaf decoctions of *Antiaris africana* Engler are applied in the traditional treatment of syphilis, sore throat and leprosy[2] , and in Senegal, used in the treatment of breast cancer. *Antiaris africana* Engle*r* is a tree about 15 to 20 m high. This tree is found in the drier forests of tropical Africa, Oceania and Southeast Asia and is widely distributed in many African countries [2,9].

Increasing evidence has revealed that phytochemicals and plant secondary metabolites have shown antitumor activities against various cancers by inducing programmed cell death. The mechanisms by which these phytochemicals trigger apoptosis could relate to their antioxidant or prooxidant properties, which involve a cascade of molecular events. ROS-mediated oxidative stress and increases in the antioxidant status in cancer cells can modulate the expression of verities of pro-apoptotic or/and anti-apoptotic regulatory proteins [10,11].

Extracts of latex, seeds, stem bark and trunk of *Antiaris africana* Engler have demonstrated pro-apoptotic effects on different cancer cells lines [12,13,14]. Moreover, the methanolic extract of the stem bark of *Antiaris africana* Engler has displayed a high toxicity and selective antitumor activity against human breast (MCF-7) and ovarian (OVCAR3) cancer cell lines [9]. However, leaf extracts of *Antiaris africana* Engler have never been investigated for their potential properties against cancer. Moreover, exploitation of the roots or barks or latex of *Antiaris africana* Engler can lead to reduction or even extinction of this plant in the long term. Herein, we report the potential cytotoxic and pro-apoptotic activity of the leaf extracts *Antiaris africana* Engler using several human cancer cell lines, as well as the results of the phytochemical investigation of these extracts by UPLC-DAD-QTOF-MS/MS.

## 2. Results

### 2.1. Pro-Apoptotic Effect of Antiaris Africana Engler Leaves Extracts on MCF-7 Cell Line

The extraction of *Antiaris africana* Engler leaves resulted in two crude extracts, EHA and Aq extracts, and six fractions, EHA-EtOAc, EHA-*n*-BuOH, EHA-Aq, Aq-EtOAc, Aq-*n*-BuOH and Aq-Aq. Firstly, the ability of both crude EHA and Aq extracts to induce cell death was examined. For this purpose, the MCF-7 cell line was incubated for 24 h with 100 µg/mL of each extract and apoptotic rates were determined using annexin V-FITC and PI assay, as mentioned in material and methods. 

As shown in Figure 1A, 88% and 85% of MCF-7 cells treated with EHA and Aq extracts were annexin V-FITC positive, respectively, whereas vehicle-treated cells showed only 12%. Here, upper and lower right quadrants were considered annexin V-FITC positive; thereby, these populations represented cells in apoptosis (Figure 1A). In addition, the same test was performed with the six fractions to investigate their apoptotic potential. The results (Figure 1B) showed that the Aq-fractions of each crude extract, EHA-Aq and Aq-Aq, were both active with 59% and 47% of cells positive to annexin V-FITC, respectively. On the other hand, the EtOAc- and *n*-BuOH- fractions of both crude extracts were not active (data not shown). Based on the observed activity, the EHA-Aq fraction was selected for further investigation.

### 2.2. Anti-Proliferative Effect of EHA-Aq on MCF-7 Cell Line

To examine the effects of different concentrations of EHA-Aq fraction on cell proliferation, MCF-7 cells were treated for 24 h with a twofold increase in concentration of EHA-Aq fraction, ranging from 12.5 µg/mL to 200 µg/mL. The plates were then scanned and cell confluence levels for each plate analyzed using image cytometry. Image cytometry generated comparable data without having to detach, thus possibly fragilize MCF-7 adherent cells. As shown in Figure 2A, EHA-Aq fraction clearly inhibited cell growth of MCF-7 cells, given that the area occupied by the cells was reduced in a concentration -dependent manner. 

The analyzed data (Figure 2B) demonstrated that a cytostatic effect of EHA-Aq fraction was noticeable already with a concentration of 12.5 µg/mL, at which point, the surface coverage by cells was reduced three times compared to the vehicle. At a concentration of 200 µg/mL, this decrease even reached 0.1%, showing the highly cytostatic to cytotoxic dose effect properties of the EHA-Aq fraction.

### 2.3. Effect of EHA-Aq Fraction on Cell Viability

In the pharmacological research, potencies and efficacy of the drugs are crucial parameters. For measuring these parameters, half-maximal response (EC_50_) is commonly used. Therefore, firstly, the EC_50_ values of EHA-Aq fraction were determined. Here, four cancer cell lines were used and the effects of different concentrations of the fraction on their viability were assessed.

After 24 h of treatment, followed by incubation with annexin V-FITC and propidium iodide (dye), cells were then analyzed by flow cytometry. Viable cells remained unstained by these dyes. The concentration of EHA-Aq fraction able to inhibit 50% of viable cells was then estimated using nonlinear regression in GraphPad prism 6.0 software (GraphPad Sofware, CA, USA). The analysis of the data showed that the EC_50_ value was 50.1 ± 0.5, 99.9 ± 0.9, 57.9 ± 7.0 and 64.6 ± 13.7 μg/mL for, respectively, THP-1, SW620, AsPC-1 and MCF-7 cell lines (Figure 3).

### 2.4. Dose-Dependent Induction of Apoptosis by EHA-Aq Fraction

There are several modalities of cell death that are classified according to morphological characteristics and measurable biochemical features [15]. To investigate whether the cytotoxic effects of the EHA-Aq fractions trigger apoptotic programmed cell death, THP-1, SW620, AsPC-1 and MCF-7 cells were treated with different concentrations of EHA-Aq fractions. Cells were evaluated for the occurrence of cell death in terms of quality and quantity. In our study, annexin-V/propidium iodide (PI) staining was used, i.e., a quantitative method that provides objective assessment of cell viability alongside an accurate incidence of apoptotic and necrotic cells in the analyzed sample. This method is based on detection of phosphatidylserine (PS) externalization in the plasma membrane surface of apoptotic cells. This is the earliest event in programmed cell death that has arisen as one of the “eat-me” signals that contributes to the recognition and removal of apoptotic bodies by phagocytosis [16]. PS is a binding site for annexin-V, while PI serves as a cell membrane non-permeable dye excluded from viable cells. Such double assay staining allows differentiation between viable cells and those in apoptosis, from early (annexin-V positive cells) to late stages (annexin-V and PI positive cells), as well as necrosis (PI only stained cells). For each cell line, the percent of annexin-V-labeled cells for each investigated concentration of *Antiaris African* Engler extract fraction was measured and the maximum apoptotic response was always verified to be around 100% with the positive control, Celastrol (50 µM). Figure 4 shows an example of cytograms obtained with different concentrations of EHA-Aq fraction (12.5–200 μg/mL) in MCF-7 cells. Positive control with full apoptotic induction due to Celastrol showed almost total induction of cell death, given that 99% of cells died due to apoptosis, whereas negative controls (cells treated with solvent) showed only 1% of dead cells. The percentages of the dead cells were plotted against the EHA-Aq fraction concentrations and the values of EC_50_ were computed using the asymmetric sigmoidal curve five-parameter logistic equation (GraphPad Prism 6 software, GraphPad Sofware, CA, USA). As seen in Figure 5, the EHA-Aq fractions showed efficient pro-apoptotic activity, inducing more than 50% cell death at a concentration of 100 μg/mL, regardless of the cell line used. The corresponding EC_50_ values calculated were 64.1 ± 1.8, 99.9 ± 2.0, 61.3 ± 1.8 and 63.5 ± 1.8 μg/mL for THP-1, SW620, AsPC-1 and MCF-7 cells, respectively. This observed variation in the response could be related to the different pathways involved in the apoptotic mechanism, depending on the phenotype of each cell line. 

### 2.5. UPLC-DAD-QTOF-MS/MS Analysis of EHA-Aq

The EHA-Aq was analyzed via UPLC-DAD-QTOF-MS/MS in negative ionization mode. The UPLC-MS chromatogram (BPC) is presented in Figure 5. Retention time, molecular formula, theoretical and experimental *m*/*z* [M-H]^−^ or [M+HCOO]^−^ values, error (Δppm), MS/MS fragmentation ions and assignment are presented in Table 1. All these compounds were identified on the bases of their exact mass, proposed molecular formula, UV and MS/MS fragmentation data and the comparison with databases available in the literature, and when possible with the data of the available authentic standard. In total, 16 compounds were tentatively identified belonging to the family of cinnamates (compounds 2–10, 15–16), flavonoid (compound 12) and cardiac glycosides (CGs) (compounds 11, 13, 14). Compound 12 has the same chemical formula, same UV, MS and MS/MS spectra, only different from rutin by retention time. Therefore, this compound was classified as a rutin isomer.

## 3. Discussion

In this study, the crude aqueous and hydroalcoholic leaves extracts of *Antiaris african* Engler with their six fractions were evaluated in vitro for their potent pro-apototic activity using annexin-V-FTIC and propidium iodine. The ability to induce cell death on the MCF-7 cell line of both crude extracts and their Aq-fractions revealed a cytotoxic effect by inducing cancer cell apoptosis, with 88%, 85%, 59% and 47% of cells positive to annexin V-FITC for EHA and Aq crude extracts, and EHA-Aq and Aq-Aq fractions, respectively, with only 12% for control cells (Figure 1A,B).

The cytotoxic activity of the stem bark [13] and latex [13,17,18] extracts of *Antiaris african* Engler has been previously demonstrated for different human cancer cell lines, including MCF-7 (breast), OVACAR-3 (ovarian), SMMC-7721 (hepatoma), DU-145 (human prostate) and NIH-H460 (lung). The decrease in cell viability for human gastric (SGC-7901) and human hepatocellular carcinoma (SMMC-7721) has also been reported, with an increase of apoptosis in human lung, colon, ovary, pancreas, prostate, uterus and stomach cancer cell lines [9,12,19].

Evaluation of the cytotoxic activity of the EHA-Aq fraction showed a dose-dependent induction of apoptosis. This fraction also induced more than 50% of cell death at a concentration of 100 μg/mL, regardless of the cancer cell lines used (THP-1, ASPC-1, SW602 and MCF-7) (Figure 5). The corresponding computed EC_50_ values were 64.1 ± 1.8, 99.9 ± 2.0, 61.3 ± 1.8 and 63.5 ± 1.8 μg/mL for THP-1, SW620, AsPC-1 and MCF-7 cells, respectively. The observed variation of EC_50_ values can be related to the different pathways involved in the apoptotic mechanism, depending on the cell line phenotype [20,21].

The phytochemical constituents of EHA-Aq fraction were determined via UPLC-DAD-QTOF-MS/MS analysis. Sixteen compounds were tentatively identified as belonging to the family of hydroxycinnamates (compounds 2–10, 15–16), flavonoids (compound 12) and cardiac glycosides (CGs) (compounds 11, 13, 14) (toxicarioside K or toxicarioside O, antiaritoxioside G, antiaroside ZC, antiaroside Z (Table 1). To our knowledge, although chlorogenic and caffeic acids have been identified in the leaf extract of *Antiaris africana* Engler [22], esters of these acids, as well as other hydroxycinnamates, such as ferulates and coumarates, were detected for the first time in the leaf extract. These compounds belong to the phenolic group, which is one of the three major classes of plant metabolites based on the biosynthetic pathway [23,24]. Hydroxycinnamates are a family of esters formed between hydroxycinnamic acid and quinic acid or other natural acids by ester bonds. Caffeic, ferulic and coumaric acids are hydroxycinnamic acids, in which one or more hydroxyl moieties of quinic acid are esterified, forming a series of positional isomers. These phenolic compounds are well-known for the antioxidant, anti-inflammatory and anti-carcinogenic effects of dietary polyphenols [25]. Our findings indicate that hydroxycinnamates are the major components of the EHA-Aq fraction (Figure 6) and may contribute to the cytotoxic and pro-apoptotic effects of the EHA-Aq fraction. Flavonoids are one of the important classes of plant-derived chemicals, with potential for cancer prevention and treatment [26,27]. Quercetin and rutin (quercetin-3-*O*-rutinoside) were previously isolated from the leaf extract of *Antiaris africana* Engler and have been reported to have significant antitumor activities against various cancers [28,29,30,31]. Therefore, the rutin isomer may be also responsible for the cytotoxic and apoptotic effect observed with the EHA-Aq fraction.

CGs are a large class of naturally occurring steroid-like compounds composed of two chemical parts, an aglycone, known as the steroidal, and a sugar. Different plants containing CGs have been used in clinics for more than 1500 years. They have been used in folk medicine as arrow poisons, abortifacients, heart tonics, emetics and diuretics, as well as in other applications [32]. CGs are reported to be present in other parts of *Antiaris africana* Engler, notably in the stem bark, the trunk bark, the latex and the seeds, from which many CGs molecules have been isolated and proved to be cytotoxic against different cancer cell lines [13,14,33]. Among these CGs, toxicarioside K has been isolated by Dong et al. [12] from the seeds of *Antiaris africana* Engler and was revealed to be cytotoxic against SMMC-7721 and SGC-7901 cancer cell lines, with IC_50_ values of 0.07 and 0.025 µg/mL, respectively [12]. Toxicarioside O, isolated also from the seeds of *Antiaris africana* Engler, has been proven to possess a strong inhibitory activity against SMMC-7721, with an IC50 value of 0.016 µM, and against human myeloid leukemia (K562) cell line, with a value of 0.053 µM [34]. Antiaroside ZC is another CG molecule that has been isolated from the bark of *Antiaris africana* Engler. Its cytotoxicity toward human NIH-H460 lung cancer cells was evaluated using MTT assays. Antiaroside ZC showed significant inhibitory effects on the proliferation of NIH-H460 cells, with an IC_50_ value of 34.18 nM, i.e., 0.24 µg/mL. Based on these results, the presence of toxicarioside K and antiaroside ZC, or other unidentified CGs, in the leaf extract of *Antiaris africana* Engler may also explain the cytotoxicity induced for THP-1, ASPC-1, SW602 and MCF-7 cells lines observed here. It has been proven that CGs exhibit anti-proliferative activity by inhibiting important signaling pathways, such as KRAS, that regulate cell growth, and are known to induce the mitochondrial apoptotic pathway via ROS-mediated oxidative stress [35,36,37]. Therefore, we expected that similar molecular mechanisms could implicate in EHA-Aq-triggered apoptotic cell death. These require further exploration.

## 4. Materials and Methods

### 4.1. Plant Names

The leaves of *Antiaris africana* Engler were collected in the department of Bignona, in the region of Ziguinchor, Senegal in May 2017. A voucher specimen (N°101) is deposited at the Laboratory of Pharmacognosy and Botanical, Dakar, Senegal. The name given to this plant: *Antiaris africana* Engler, synonym of *Antiaris toxicaria* Var africana Scott Elliot ex A. chev is verified on the site http://www.theplantlist.org/ (accessed on 31 March 2022). In Senegal, the plant is commonly called “Buffo”, “Bafor” and “Tufu”, among the “Djola” speaking people of southern Senegal, “Mbayo” and “Nget yana” among the “Serere” speaking people of Eastern Senegal, “Kan” among the “wolof” in the western end of Africa, on the narrow Cape Verde peninsula. In other African countries, notably the Ivory Coast, the plant is called “Gouho” by the “Fon” ethnic group and, in Guinea Conakry, “Cili” by the Malinké ethnic group [38]. *Antiaris africana* Engler is a big tree with heavy flat crown and blotchy grey and white bark. The small greenish-white flowers give rise to red, velvety fruits [39].

### 4.2. Materials and Reagents

The samples were air-dried for three weeks, and then powdered by a cryo-grinder. All solvents were of analytical grade. Ethanol (EtOH) was purchased from Merck Sigma Aldrich (Darmstadt, Germany). Ethyl acetate (EtOAc) and *n*-butanol (*n*-BuOH) were purchased from VWR (Rosny-sous-Bois, France). Milli-Q water (18.2 MΩ) was generated by Millipore synergy system (Molsheim, France). Phosphate buffer solution (PBS) was prepared as follows: 137 mM sodium chloride (NaCl), 2.7 mM potassium chloride (KCl), 10 mM disodium hydrogen phosphate dihydrate (Na_2_HPO_4,_ 2H_2_O) and 1.76 mM potassium dihydrogen phosphate (KH_2_PO_4_).

### 4.3. Sample Preparation

The leaf powder of *Antiaris africana* Engler was extracted using either ultrasonic extraction with 70% EtOH or maceration with water. Each extract was suspended in water and then sequentially fractionated with EtOAc and *n*-BuOH to obtain three fractions. 

#### 4.3.1. Ultrasonic Extraction

Firstly, 2.0 g of leaves powder was mixed with 25 mL 70% EtOH, and sonicated for 15 min in a Fisherbrand 15,051 ultrasonic bath (Fisher Scientific, Loughborough, UK, 37 kHz, 280 W). After centrifugation at 5000× *g* rpm for 10 min, the supernatant was recovered. The residue was re-extracted under the same conditions two times. The supernatants were combined. After elimination of EtOH by rotary evaporation, the sample was freeze-dried. The obtained dry crude extract (EHA crude extract) was suspended in water, and then sequentially partitioned with EtOAc and *n*-BuOH to obtain EtOAc-, *n*-BuOH- and water fractions, respectively. After elimination of the solvent, three resulting dry fractions were obtained and named, respectively, EHA-EtOAc, EHA-*n*-BuOH and EHA-Aq, and stored at −20 °C.

#### 4.3.2. Maceration Extraction

Firstly, 2.0 g of leaves powder was macerated in 25 mL of water for 48 h under magnetic agitation at room temperature. After centrifugation at 5000× *g* rpm for 10 min, the supernatant was recovered. The residue was re-extracted under the same conditions two times. The accumulated supernatants were treated in the same way as described for the ultrasonic extraction to obtain the crude aqueous extract (crude Aq extract). The three resulting fractions were named, respectively, Aq-EtOAc, Aq-*n*-BuOH and Aq-Aq, and stored at −20 °C.

### 4.4. Cell Culture

All human cell lines purchased from ATCC (LGC Standards, Molsheim, France) Human mammary (MCF-7, ATCC^®^ HTB-22), pancreatic (AsPC-1, ATCC^®^ CRL-1682) and colon (SW-620, ATCC^®^ CCL-227^™^) adenocarcinoma cell lines were maintained in DMEM high-glucose medium (Dominique Dutscher, 67,172 Brumath, France, Cat No L0102-500), while the human acute monocytic leukemia cell line (THP-1, ATCC^®^ TIB-202) was maintained in RPMI-1640 medium (ATCC^®^ 30-2001™, LGC Standards, Molsheim, France), supplemented with 10% (*v*/*v*) heat-inactivated fetal bovine serum (FBS, Life Technologies, Paisley, UK, Cat No 10270-106) and 1% (*v*/*v*) penicillin-streptomycin (10,000 units/mL and 10,000 µg/mL, Life Technologies, Paisley, UK, Cat No 15140-122). Cells were kept at 37 °C in a humidified atmosphere containing 5% (*v*/*v*) CO_2_ during their exponential growing phase and in the course of incubation with investigated compounds. Before confluence, adherent cells were trypsinized and sub-cultured twice a week.

Investigated extracts were initially dissolved in dimethyl sulfoxide (DMSO) in a concentrated stock solution. Further dilutions to the experimental concentrations applied on the cells have been done in RPMI-1640 or DMEM media prior to each experiment, thus the final concentration of DMSO on treated cells was 0.5% (*v*/*v*) for the most applied concentrations.

### 4.5. Cytomic Analysis

#### 4.5.1. Cytotoxicity Assay by Microcapillary Flow Cytometry

To study the cytotoxic effects of our extracts, viability and apoptosis rates were determined. Here, annexinV-FITC and propidium iodide, characteristic markers for cell apoptosis and necrosis, were used. In brief, cells at density of 2 × 10^4^ cells/mL were seeded in 96-well flat bottom plates (Corning^®^ Costar^®^, Cat. No. CLS3596). Adherent cells were left overnight to settle prior to treatment. Different extract fractions of *Antiaris africana* Engler leaves were added to generate 5 final concentrations, ranging from 12.5 to 200 µg/mL. As controls, non-treated cells, vehicle-treated cells and cells treated with 50 µM Celastrol (as a potent inducer of apoptosis) (Enzo Life Sciences, Cat. No. ALX-350-332-M025) were used. Plates were then incubated for 24 h at 37 °C under 5% CO_2_. Cells in suspension were required for flow cytometric analysis. Therefore, adherent cells were collected by trypsin-EDTA, and then resuspended in its original incubation medium. Cells in suspension were then incubated with 3 µL of annexinV-FITC (ImmunoTools GmbH, Friesoythe, Germany, Cat No 31490013) and 2 µL of 1 mg/mL propidium iodide (PI, Miltenyi Biotec Inc., Auburn, AL, USA, Cat No 130-093-233) for 15 min in the dark. Death cells were assessed by capillary flow cytometry (Guava EasyCyte Plus, Guava/Luminex, Santa Clara, CA, USA). A minimum of 5000 cells were acquired per sample and analyzed on the InCyte software (Guava/Luminex, Santa Clara, CA, USA). Set up dot plots with log scale were analyzed to determine non-labeled cells (viable cells) or annexinV-FITC-positive cells (apoptotic cells) and PI-positive cells (necrotic cells). It should be mentioned that the percentage of viable and apoptotic cells was used to compute the EC50.

#### 4.5.2. Cell Growth Assay by Image Cytometry

Dose-dependent effects of EHA-Aq fraction on the growth of MCF-7 cells were evaluated using Celigo imaging cytometer (Nexcelom Bioscience LLC, Lawrence, MA, USA). Cells were prepared and treated as mentioned above (Section 4.4). After 24 h extract incubation, plates were scanned using the confluence application and images were analyzed. The number of cells was then compared to the occupied area.

### 4.6. Phytochemical Analysis by UPLC-DAD-QTOF-MS/MS

The EHA-Aq fraction was analyzed using a high-resolution Bruker micrOTOF-Q II (Bruker, Karlsruhe, Germany) hyphenated with a Waters Acquity UPLC system (Waters, Guyancourt, France). Chromatographic separation was achieved by C18-PFP column (250 × 4.6 mm, 3 µm, ACE, Aberdeen, Scotland). The mobile phase was composed of 0.05% formic acid in water (A) and 0.05% formic acid in ACN (B) and delivered at 1.0 mL/min with a gradient elution: 5–10–10–15–15–100 B% (0–15–25–30–40–68 min). The column was then washed and re-equilibrated for another injection. The injection volume was 10 µL. UV spectra of all compounds were recorded between 200–500 nm. The mass spectrometry (MS) was operated in negative electrospray ionization mode, with the mass range from *m*/*z* 100 to *m*/*z* 1500. Nitrogen was used as drying gas (9.0 L/min at 200 °C), nebulizing gas (40.6 psi) and collision gas. The capillary voltage and the end plate offset were set to 4500 V and 500 V, respectively. The collision energy was set to 30 eV for MS/MS experiments. The operating software was micrOTOF 3.0 combined with Hystar 3.2 and data analysis software was Compass 1.3 (Bruker, Karlsruhe, Germany). 

### 4.7. Statistical Analysis

GraphPad Prism 6 was performed to determine EC_50_ values using the nonlinear regression curve fit for inhibitory dose–response. The data are presented as mean  ± SD.

## 5. Conclusions

The present work provides, for the first time, the preliminary evidence to support the use of *Antiaris africana* Engler leaves in the treatment of different diseases and cancers, especially in Africa. Our results show that the leaf extract of *Antiaris africana* Engler efficiently induces apoptotic cell death in breast, colon, pancreatic and leukemic cancer cell lines in a dose-dependent manner. The analysis of the phytochemical constituents of the extract using UPLC-DAD-QTOF-MS/MS revealed the presence of different families of compounds, with hydroxycinnamates as main components, one rutin isomer (quercetin glycoside) and three cardiac glycosides. As far as we know, hydroxycinnamates and the rutin isomer were detected for the first time in *Antiaris africana* Engler, and they are all phenolic compounds. As known, different phenolic compounds have been reported to have antioxidant, anti-inflammatory and anti-carcinogenic properties. As for cardiac glycosides, many of them have been isolated previously from seeds and bark of *Antiaris africana* Engler, and have been reported to be cytotoxic for different cancer cell lines. Therefore, our findings suggest that the leaves of *Antiaris africana* Engler may be a potential source of anticancer compounds, although much work remains to be done. Further studies, including bioguided isolation of the bioactive compounds and individual quantitative bioassays, followed by chemical structure elucidation of the active molecules by high-resolution mass spectrometry (HRMS) and full nuclear magnetic resonance (NMR) analysis, are necessary. Furthermore, possible mechanisms of action should also be investigated. 

## Figures and Tables

**Figure 1 molecules-27-07723-f001:**
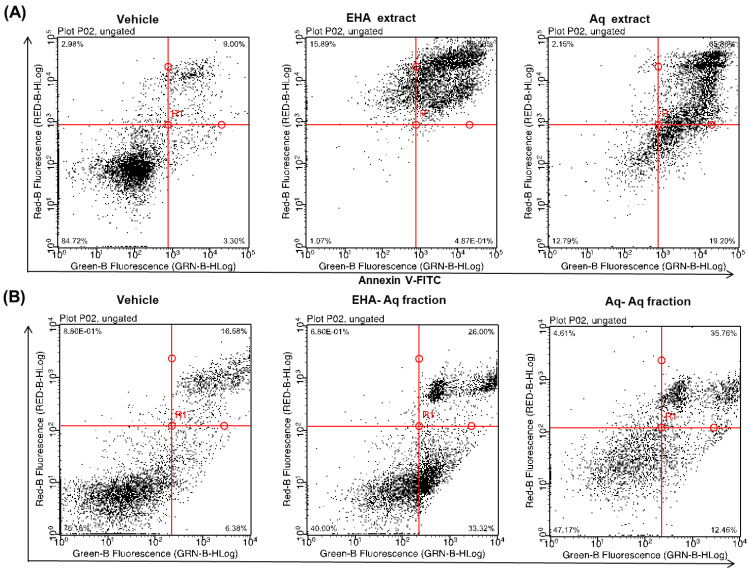
Apoptosis induction by crude extracts (EHA and Aq, 100 μg/mL) after 24 h (**A**), as well as the corresponding aqueous fractions (EHA-Aq and Aq-Aq, 100 µg/mL) (**B**) of *Antiaris africana* Engler leaf extracts in MCF-7 cells compared with the control test. Lower-left quadrant—viable cells; lower- right—early apoptotic cells; upper-right—late apoptotic cells; upper-left—necrotic cells.

**Figure 2 molecules-27-07723-f002:**
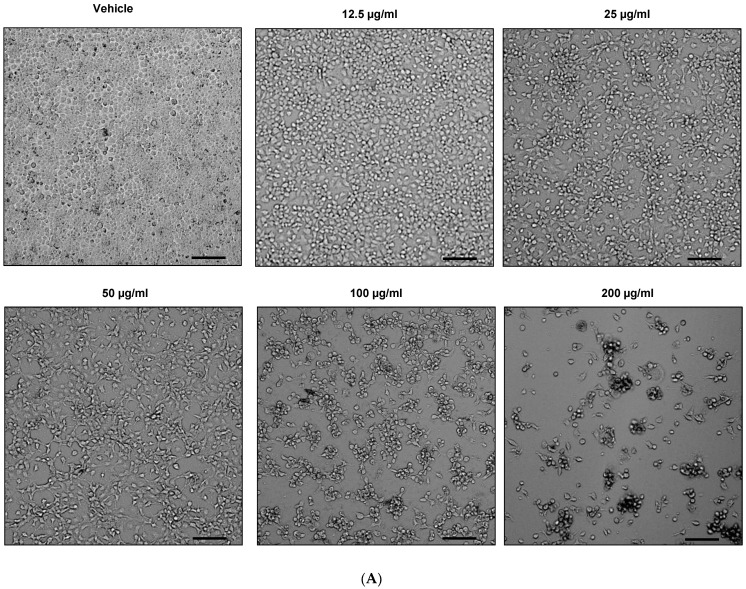
Dose-dependent effects of EHA-Aq fraction on MCF-7 cell proliferation. (**A**) Representative bright-field images of cell confluence captured by Celigo cytometer after 24 h of treatment. (**B**) Bar graphs show nice correlation between surface area occupied by the cells and number of cells after confluence analysis by Celigo software (Cyntellect Inc, San Diego, CA, USA).

**Figure 3 molecules-27-07723-f003:**
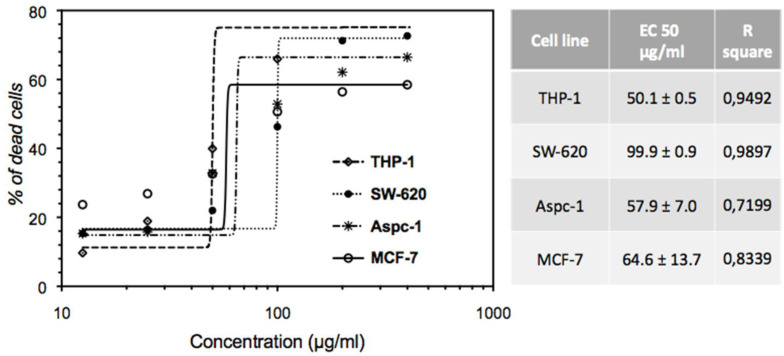
Dose-dependent effects of EHA-Aq fraction on cell viability in different phenotypes of human cancerous cell lines. The cells were treated with various concentrations for 24 h, and then cell death induction was analyzed by flow cytometry. EC_50_ values were calculated using GraphPad Prism 6 software.

**Figure 4 molecules-27-07723-f004:**
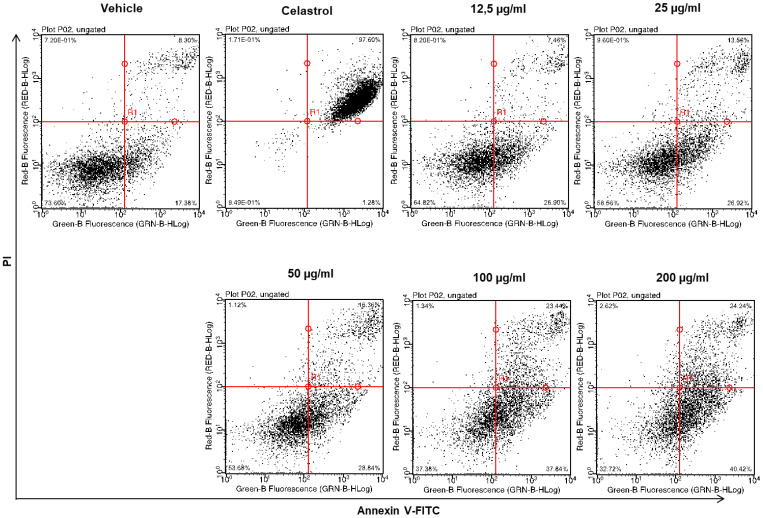
Dose-dependent induction of apoptosis by EHA-Aq fraction (12.5–200 μg/mL) on MCF-7 cell line after 24 h of treatment. Celastrol inducing a 99% apoptotic response was used to delimit the quadrant determining apoptotic states.

**Figure 5 molecules-27-07723-f005:**
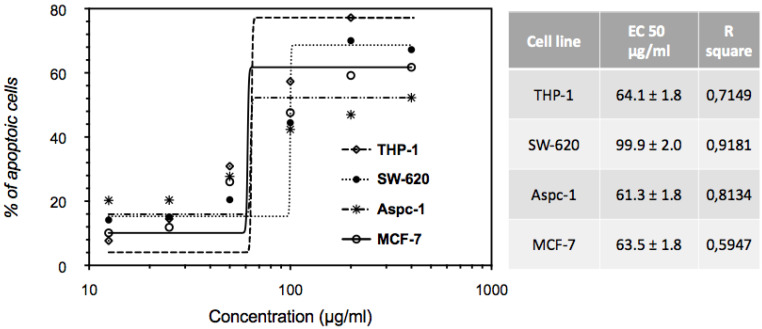
Dose-dependent induction of apoptosis by EHA-Aq fraction (12.5–200 μg/mL) in different phenotypes of human cancerous cell lines. After 24 h of treatment, apoptosis induction was analyzed by flow cytometry. Percentages of apoptotic cells were used to calculate EC_50_ values using GraphPad Prism 6 software.

**Figure 6 molecules-27-07723-f006:**
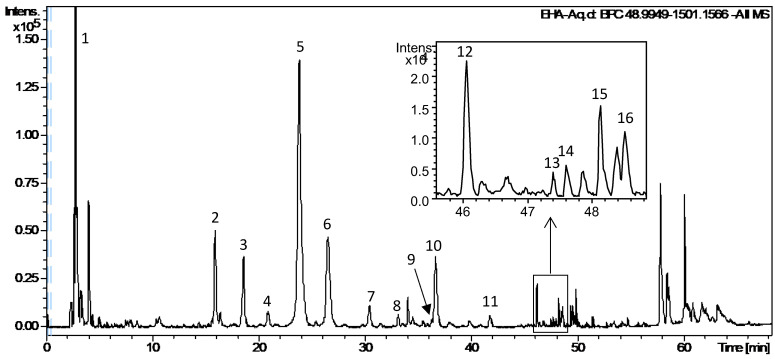
UPLC-MS chromatogram (BPC) of EHA-Aq fraction of *Antiaris africana* Engler leaves. The marked number is consistent with that of Table 1.

**Table 1 molecules-27-07723-t001:** UPLC-DAD-QTOF-MS/MS data of identified compounds in EHA-Aq fraction.

Peak N°	R_t_ (min)	UV (nm)	Formula (*m*/*z*) [M-H]^−^	Theo. (*m*/*z*)	Exp. (*m*/*z*)	Error (ppm)	Fragmentation (*m*/*z*)	Identification
1	2.75	-	C_7_H_11_O_6_	191.0561	191.0556	2.7	-	Quinic acid
2	15.83	325, 300sh	C_16_H_17_O_9_	353.0878	353.0867	3.2	191.0556; 179.0344; 135.0432	Chlorogenic acid isomer
3	18.50	342, 312sh, 290sh, 252	C_32_H_33_O_18_	705.1672	705.1651	3.0	513.1023; 339.0496; 321.0418; 229.0129; 191.0553	Caffeoylquinic acid derivative (biphenyl type) ^a^
4	20.80	340, 321, 285sh, 252sh	C_32_H_33_O_18_	705.1672	705.1662	1.5	513.1012; 339.0493; 321.0418; 191.0541	Caffeoylquinic acid derivative (biphenyl type) ^a^
5	23.73	327, 300sh	C_16_H_17_O_9_	353.0878	353.0876	0.5	191.0557	Chlorogenic acid
6	26.44	325, 300sh	C_16_H_17_O_9_	353.0878	353.0868	2.8	191.0558; 179.0333; 173.0450, 135.0442	Chlorogenic acid isomer
7	30.30	320	C_16_H_17_O_9_	353.0878	353.0870	2.3	191.0557	Chlorogenic acid isomer
8	33.03	312, 290sh	C_16_H_17_O_8_	337.0929	337.0922	2.0	191.0566; 163.0395	Coumaroylquinic acid isomer
9	36.20	310sh, 280sh	C_16_H_17_O_8_	337.0929	337.0923	1.6	191.0555	Coumaroylquinic acid isomer
10	36.56	325, 300sh	C_17_H_19_O_9_	367.1035	367.1019	4.3	191.0561	Feruoylquinic acid
11	41.65	-	C_30_H_45_O_13_[M+HCOO]^−^	613.2866	613.2828	6.1	567.2789; 405.2231	Toxicarioside K or Toxicarioside O
12	46.08	342	C_27_H_29_O_16_	609.1461	609.1433	4.6	301.0328; 300.0266	Rutin isomer ^b^
13	47.40	-	C_30_H_45_O_12_ [M+HCOO]^−^	597.2917	597.2889	4.5	551.2824; 389.2338	Antiaritoxioside G
14	47.60	-	C_36_H_53_O_16_[M+HCOO]^−^	741.3339	741.3299	5.4	695.3246 ; 533.2776 ; 385.1990 ; 161.0431	Antiaroside ZC
15	48.16	325, 290sh	C_25_H_23_O_12_	515.1195	515.1169	5.1	353.0841; 191.0564; 179.0339	Caffeoylquinic acid derivative ^c^
16	48.44	327, 290sh	C_25_H_23_O_12_	515.1195	515.1167	5.5	353.0846; 191.0549; 179.0335	Caffeoylquinic acid derivative ^c^

^a^: The compound contains two caffeic acids linked by a biphenyl bond and two quinic acids. One of the quinic acids is linked by ester formation between the −COOH group of caffeic acid and one of the four −OH groups of quinic acid. The second quinic acid is linked by a glycosyl bond formed between one of the four −OH groups of the second quinic acid and one of the two −OH groups of the second caffeic acid. ^b^: The compound is different from rutin due to its aglycone, which is an isomer of quercetin. ^c^: The compound contains two caffeic acids and one quinic acid. One of the caffeic acids is linked by ester formation between the −COOH group of caffeic acid and one of the four −OH groups of quinic acid, and the second is linked by a glycosyl bond formed between one of its two −OH groups and one of the four −OH groups of quinic acid.

## Data Availability

Not applicable.

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
