# Peer review of "Cytotoxic and Pro-Apoptotic Effects of Leaves Extract of Antiaris africana Engler (Moraceae)"

_molecules, 2022, doi:10.3390/molecules27227723_

Round 1
Reviewer 1 Report
Dear Authors,
The MS “Cytotoxic and pro-apoptotic effects of leaves extract of Antiaris africana Engler (Moraceae)” is original research topic which support plant biodiversity research and use native plants for plant-based drugs development. Especially important in such study research of plant extracts of different parts. The MS can be accepted for publishing after minor revision.
In Abstract part would be good to add few words about identified major biochemical compounds. After sentence “The phytochemical components of the active extract are investigated by UHPLC-DAD-HRMS-MS” . It is good to say that was identified chlorogenic acid and its derivatives, Caffeoylquinic acid derivatives and rutin.
Introduction part would be good to say about role of plant biodiversity due presence of huge amount specified secondary metabolites in the development plant-based drugs. To discover presence of some metabolites used modern techniques but more important to study healthy effects of plant extracts.
It would be good to add this sentence after L41-42 Introduction part “In Africa, traditional medicine is sometimes the only source of affordable and accessible care, especially for the poorest patients. This traditional African medicine mainly uses herbal therapies “
Please, see next references bellow
Sytar O, Bruckova K, Hunkova E, Zivcak M, Konate K, Brestic M. The application of multiplex fluorimetric sensor for the analysis of flavonoids content in the medicinal herbs family Asteraceae, Lamiaceae, Rosaceae. Biol Res. 2015 Jan 16;48(1):5. doi: 10.1186/0717-6287-48-5.
Twaij, B.M.; Hasan, M.N. Bioactive Secondary Metabolites from Plant Sources: Types, Synthesis, and Their Therapeutic Uses. Int. J. Plant Biol. 2022, 13, 4-14. https://doi.org/10.3390/ijpb13010003
In Conclusion part its would be good to say about future perspectives that it is important to make further studies about quantitative analysis of identified secondary metabolites to find possible relation of their presence with antioxidant and other studied healthy effects.
Reviewer 2 Report
This is an interesting and sound presentation. It is well-written with some minor grammar issues. I believe it contributes to the overall collection of potential anticancer agents.
I would like to see a brief discussion of apoptosis pathways in the Introduction, perhaps with a diagram. This can be discussed in the light of other cancer therapies. A similar review of mechanism should then be integrated into the discussion with reference to the potential uses for the extracts.
I was confused in reading by the references to multiple controls - cells only, vehicle or apoptotic-inducer. This needs to be better clarified in each figure legend and in labeling the graphs.
Please note the time course for the individual experiments by listing these in the figure legends.
PLease include the mathematical formula for calculating %. Note the starting viability in each figure legend. This can be a blanket statement that all were >95%.
Some specific corrections:
Table I capitalize "p" henotypes.
Table II correct formatting in table on lines 11, 13, and 14
In discussion there is reference to ....depending the cell line phenotype. I think you are referring to lineage here. That is to say that lineage implies tissue of origin and differentiation which likely determines the cell's susceptibility to apoptosis. This line is also missing a preposition (to).
Reviewer 3 Report
Thiam et al. investigated a research titled “Cytotoxic and pro-apoptotic effects of leaves extract of Antiaris africana Engler (Moraceae)”. Although the research is scientifically sound, it must be revised to solve some important issues before it may be considered for publication in the Molecules Journal.
1. Abstract: It is essential to rewrite the objective in Lines 12–13, "The present study aimed to investigate the anticancer potential of Antiaris africana Engler using several human cancer cell lines". All small numbers in the manuscript must be stated in verbal form (For example 3 parts – three parts, 3 cardiac glycosides – three cardiac glycosides, 2 crude extracts - two crude extracts, 6 fractions – six fractions etc.). The conclusion part should contain more sentences.
2. Introduction: Include a paragraph outlining the role of natural products and medicinal plants that have been shown anticancer properties. The research questions might be more clearly stated. The statement (This is why in this study....) on lines 57–60 is poorly written. I imagine that after reading such an objective, the readers will be confused. Need to write it newly.
3. Results: Line 76, no data shown. This information, in my opinion, is equally crucial for readers to learn about the outcomes of other fractions. Therefore, include it to the manuscript. What does "little better activity" in line 77 mean? The authors should improve on their writing. English native speakers should review the entire manuscript. Legend for Figure 2: Replace A) with a) and B) with b). Similarly Figure or Fig. When mentioning such things in the manuscript, be consistent. Table 1 appears to have been duplicated. The format needs to be changed and converted back to the original Table format, and references are required for each component included in the Table. With the exception of the aforementioned modifications, the results section's format is generally appropriate.
4. Discussion: It is best to avoid repeating the results in the discussion section. Principal relationships, and generalizations backed by the findings are all important components of a solid discussion. The discussion section could also be improved by adding the most recent research on Antiaris africana Engler and other related plants' anticancer properties, which will support the current findings.
5. Materials and Methods: Include the reference number for the plant voucher specimen. Add more references to support the procedure used. Aside from that, I'm really happy with the materials and methods portion.
6. Conclusion: Again, some repetition of results in numerical values. This must be avoided in the conclusion. Novelty of the work should be supplemented by the author (in the conclusion section). This section should be supported by the results/insights. Conclusively, it will confer a distinct idea of the study. Author should stress the significance of the study.
7. References: References in the introduction section are too outdated. Authors are required to provide some recent sources when discussing the importance of natural products for cancer treatments as well as the global scenario of cancer patients today (https://doi.org/10.2147%2FBCTT.S316667; https://doi.org/10.1016/j.sjbs.2021.07.046).
Round 2
Reviewer 3 Report
The author responded to all of my inquiries and revised the manuscript content accordingly. As a result, I suggest that it be taken into consideration for publication in Molecules Journal in its current form.